# INSBANK: EVOLVING INSTRUCTION SUBSET FOR ONGOING ALIGNMENT

## ABSTRACT

Pre-trained large language models (LLMs) typically undergo instruction fine-tuning to improve alignment. Recent research highlights that the quality and diversity of instruction data are more critical than data quantity, prompting the selection of diverse, high-quality instruction subsets to reduce training costs. However, how to evolve these selected subsets alongside the development of new instruction data remains insufficiently explored. To achieve LLMs' ongoing alignment, we introduce Instruction Bank (**InsBank**), a continuously updated repository that integrates the latest valuable instructional data. We further propose Progressive Instruction Bank Evolution (**PIBE**), a novel framework designed to evolve InsBank effectively and efficiently over time. It firstly employs a gradual data selection strategy to maintain long-term efficiency, utilizing a representation-based diversity score that captures relationships between data points and retains historical information for comprehensive diversity evaluation. This also allows for flexible combination of diversity and quality scores during data selection and ranking. Extensive experiments demonstrate that PIBE significantly outperforms baseline methods in evolving InsBank. Additionally, PIBE enables users to flexibly extract smaller subsets based on their specific budget.

## 1 INTRODUCTION

Instruction fine-tuning are widely adopted to refine pre-trained LLMs to accurately understand human instructions and provide precise, pertinent and harmless responses (Longpre et al., 2023; Qin et al., 2024a). LIMA (Zhou et al., 2023) has proved that the quality and diversity of instruction data are significantly more critical than its sheer quantity for training, motivating recent efforts in instruction data selection to reduce unnecessary training costs by eliminating low-quality and redundant data (Qin et al., 2024a). However, how to evolve the selected instruction subset in parallel with the development of the instruction data remains underexplored.

Specifically, with the continuous emergence of new instruction datasets, it becomes necessary to regularly update the instruction subset to incorporate the latest high-quality instruction data, ensuring ongoing improvements in the alignment capabilities of LLMs. Simultaneously, the subset size must be controlled to avoid excessive growth that could lead to increased training costs. To address these practical challenges, we propose a novel concept termed **InsBank** (Instruction Bank). InsBank is initially established through a selective process applied to current available instruction data. As new instruction datasets is proposed, the bank evolves by selecting new data while phasing out an equivalent amount of older data, thereby maintaining an optimized instruction subset. Figure 1 illustrates this pipeline. Additionally, the data in InsBank should be ordered, allowing users to efficiently extract a smaller subset tailored to their specific training budget based on the ranking.

Quality can be easily scored through manual annotation or model annotation. However, regarding diversity, global measurement between data is required, which demands significant storage and computational costs. Naively, the evolution of subset can be achieved by data re-selection across all available data at each evolution iteration. However, the vast volume of instruction data (Qin et al., 2024a) and its rapid development (Longpre et al., 2023; Wang et al., 2023; Xu et al., 2023) make the costs in this manner unacceptable. Additionally, since the data in InsBank must be ordered, each instruction requires an individual score for ranking purposes. Existing methods, however, struggle to properly represent and combine diversity and quality scores.

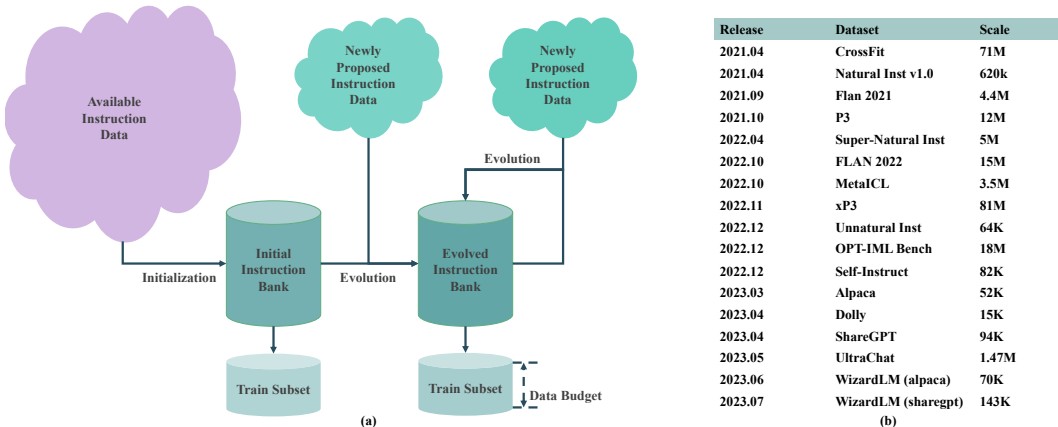

| Release | Dataset | Scale |
|---|---|---|
| 2021.04 | CrossFit | 71M |
| 2021.04 | Natural Inst v1.0 | 620k |
| 2021.09 | Flan 2021 | 4.4M |
| 2021.10 | P3 | 12M |
| 2022.04 | Super-Natural Inst | 5M |
| 2022.10 | FLAN 2022 | 15M |
| 2022.10 | MetaICL | 3.5M |
| 2022.11 | xP3 | 81M |
| 2022.12 | Unnatural Inst | 64K |
| 2022.12 | OPT-IML Bench | 18M |
| 2022.12 | Self-Instruct | 82K |
| 2023.03 | Alpaca | 52K |
| 2023.04 | Dolly | 15K |
| 2023.04 | ShareGPT | 94K |
| 2023.05 | UltraChat | 1.47M |
| 2023.06 | WizardLM (alpaca) | 70K |
| 2023.07 | WizardLM (sharegpt) | 143K |

Figure 1: (a) Pipeline of instruction bank evolution. It is initialized by data selection on all current available instruction data, and it will evolve itself as long as new instruction data are proposed. Besides, a smaller training subset can be obtained from InsBank according to the user training budget. (b) The timeline of the release of instruction datasets clearly illustrates the rapid pace of their development.

To address these challenges, we propose Progressive Instruction Bank Evolution (**PIBE**) to continuously and efficiently select the current optimal instruction subset. Firstly, it employs a gradual manner of selection to evolve InsBank, ensuring long-term efficiency. Unlike the naive method, we significantly reduce costs by excluding the large volume of data already filtered out in previous iterations, and only require the newly proposed and current instruction data in InsBank to update it. Based on this foundation, since discarded data can influence the diversity score of new data, preserving historical information throughout the evolution process is essential. Previous diversity-driven data selection methods (Liu et al., 2024; Wu et al., 2023) can be classified into two categories: k-nearest neighbor (k-NN) (Dong et al., 2011) and geometry-based coreset sampling (Guo et al., 2022). However, both of them rely solely on information from a few surrounding points, making it difficult to record and utilize the rich information of historically eliminated points. For the same reason that they cannot capture the distribution relationships between global points, they also fail to provide robust individual diversity scores for ranking. Inspired by Affinity Propagation (Frey & Dueck, 2007), we pass information between two types of messages, where multiple iterations and the preservation of historical distribution through similarity propagation enable the diversity score to capture relationships between samples, quantifying how well an instruction represents others while remaining irreplaceable. Furthermore, existing data selection methods either focus on quality or diversity alone (Chen et al., 2024), or consider them sequentially (Liu et al., 2024), failing to address both simultaneously with equal consideration. Our diversity score can be seamlessly combined with the quality score, enabling comprehensive and flexible instruction selection and ranking.

We simulate the process of instruction set development with five datasets: Self-Instruct (Wang et al., 2023), Alpaca (Taori et al., 2023), Dolly (Conover et al., 2023), ShareGPT (Chiang et al., 2023) and WizardLM (Xu et al., 2023). We apply PIBE on them and fine-tune the Llama3-8B (AI@Meta, 2024) model on the InsBank obtained from each evolution iteration. We evaluate the fine-tuned models on AlpacaEval (Li et al., 2023b) and MT-Bench (Zheng et al., 2023). Experimental results show that PIBE outperforms the baseline data selection methods and successfully evolves the instruction bank in parallel with the development of instruction sets. Besides, analysis on ordering of InsBank indicates that users can flexibly select a smaller subset based on their budget.

- We introduce a new concept called InsBank to evolve the selected instruction subset in parallel with the development of the instruction data to achieve ongoing alignment.

- We propose Progressive Instruction Bank Evolution, which efficiently obtains the optimal current instruction subset, utilizing a highly representative diversity score with memory capabilities and allowing flexible combination with quality scores.

- Extensive experiments show that PIBE not only significantly outperforms baseline methods in evolving InsBank, but also allows users to flexibly extract smaller subsets tailored to their specific budgets.

## 2 PRELIMINARIES

### 2.1 INSTRUCTION DATA SELECTION PROBLEM

Following Liu et al. (2024), given a collection of instruction data, $\mathbb{X} = \{x_1, x_2, ..., x_n\}$ where $x_i$ is an individual instruction-response pair, data selection selects an instruction subset $\mathbb{P}_\pi^m$ of size $m$ from $\mathbb{X}$, where $\pi$ is the data selection strategy. Denote the IFT performance evaluation function for $\pi$ as $Q$, the optimal data selection strategy $\pi^*$ with subset size $m$ satisfies:

$$\pi^* = \arg\max_\pi Q(\mathbb{P}_\pi^m) \tag{1}$$

### 2.2 SELECTION METRICS

Many studies (Liu et al., 2024; Qin et al., 2024a) have highlighted that the effectiveness of instruction set selection depends on both quality and diversity. In line with this understanding, we also focus these two aspects in this paper:

**Quality** The quality of instruction data primarily refers to the accuracy and rationality which estimate the consistency and coherence of the instruction context, as well as whether the response accurately corresponds to the instructions (Qin et al., 2024a). Leveraging the strong power of ChatGPT models (i.e. GPT-3.5-turbo and GPT-4 (OpenAI, 2023)), recent works typically employ a GPT-model to annotate the quality of instruction data with a specifically designed prompt (Chen et al., 2024). We adopt the quality evaluation model and method from DEITA (Liu et al., 2024) in this work.

**Diversity** The diversity of dataset is critical to the generalization ability of the trained model (Qin et al., 2024a). There are currently two major approaches to measuring diversity.: k-nearest neighbor (k-NN) (Dong et al., 2011) and geometry-based coreset sampling (Guo et al., 2022). The former measures sample's diversity by its distance (or similarity) to its $j$-th k-nearest neighbor (k-NN) with the help of text embeddings as shown in Eq. 2:

$$kNN_i^j = d(e(x_i), e(N_j(x_i))) \tag{2}$$

where $N_j(x_i)$ denotes the $j$-th closest neighbor of $x_i$ in the embedding space projected by $e(\cdot)$, and $d(\cdot, \cdot)$ calculates the euclidean distance between $x_i$ and $N_j(x_i)$. The latter is to find the most informative-and-diverse subset that represents the entire dataset the most through controlling the minimum distance between any two samples for subset selection (Guo et al., 2022; Sener & Savarese, 2018). However, both methods rely solely on local information from nearby points, making it difficult to capture the global distribution relationships or utilize historically eliminated points, resulting in inadequate individual diversity scores for subset evalution.

### 2.3 AFFINITY PROPAGATION

Affinity Propagation (AP) (Frey & Dueck, 2007) is a clustering algorithm that leverages message-passing to uncover the global distribution of the data. It identifies exemplars by iteratively transmitting two kinds of messages between data points:

- **Responsibility ($R[i, k]$)** This message sent from point $i$ to point $k$ represents how suitable point $k$ is to serve as the exemplar for point $i$.

- **Availability ($A[i, k]$)** This message sent from point $k$ to point $i$ represents how appropriate it would be for point $i$ to choose point $k$ as its exemplar, taking into account the current responsibilities sent from other points to $k$.

The messages are updated iteratively based on the following rules:

$$R[i,k] \leftarrow S[i,k] - \max_{k' \neq k} \{A(i,k') + S(i,k')\},$$

$$A[i,k] \leftarrow \min \left\{ 0, R[k,k] + \sum_{i' \notin \{i,k\}} \max\{0, R[i',k]\} \right\}, \tag{3}$$

$$A[k,k] \leftarrow \sum_{i' \neq k} \max\{0, R[i',k]\}.$$

Here, $S[i,k]$ represents the similarity between point $i$ and point $k$ where $i \neq k$. And $S[k,k]$ is filled by the predefined preference value which represents the preference for sample $i$ as a cluster center.

In this paper, we take the results of message-passing as the results of exemplar election, where the $i$-th row of $R$ represents the willingness of $x_i$ to be represented by other data, and the $k$-th column of $A$ represents the willingness of each data to be represented by $x_k$. Utilizing $A$ and $R$, we calculate the representation score of each data (i.e. the individual diversity score) for the purpose of selecting a group of data that best represent all available instruction data.

## 3 PROGRESSIVE INSTRUCTION BANK EVOLUTION

In this section, we provide a detailed explanation of our proposed method, PIBE, which comprises four core elements as shown in Figure 2: the gradual manner of evolution, the flow of historical information across evolution rounds, individual representation scoring for diversity evaluation, and the integration of quality and diversity scores for data selection and ranking.

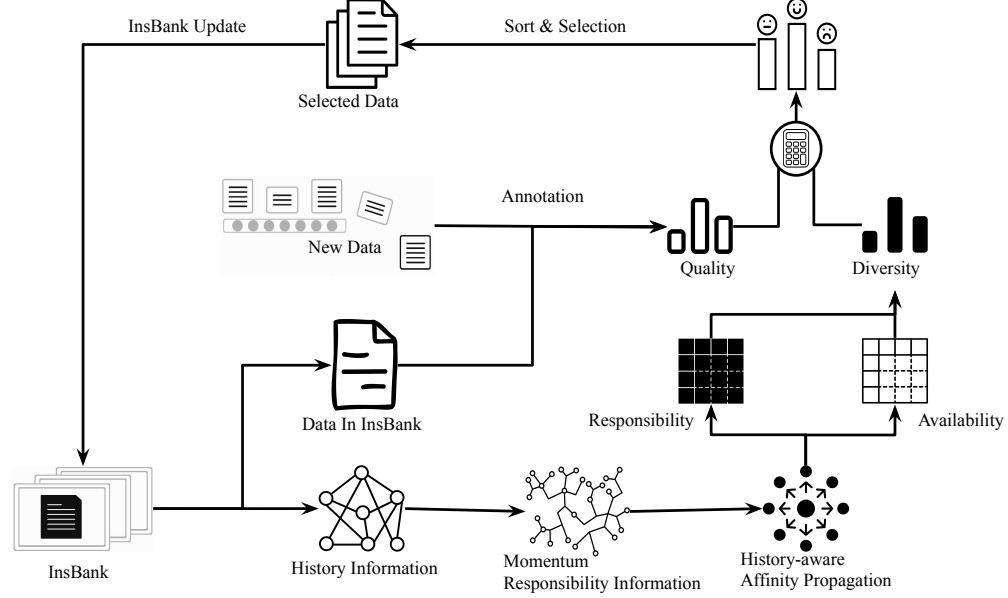

Figure 2: The detailed pipeline of PIBE.

### 3.1 GRADUAL EVOLUTION FORMULATION

In this work, we propose the instruction subset evolution task to build the InsBank. Denoting current available instruction data as $\mathbb{X}_0$, the instruction bank $\mathbb{B}_0^m$ of size $m$ is initialized through data selection which can be presented as $\mathbb{B}_0^m = \pi(\mathbb{X}_0)$. Then, when new instruction dataset $\mathbb{X}_1$ is proposed, $\mathbb{B}_0^m$ will evolve itself to adapt to changes in data distribution. The naive manner of InsBank evolution can be represented as $\mathbb{B}_1^m = \pi(\mathbb{X}_0, \mathbb{X}_1)$ which can be extended to $\mathbb{B}_{t+1}^m = \pi(\mathbb{X}_0, ..., \mathbb{X}_t, \mathbb{X}_{t+1})$ for

future evolution. However, this manner requires substantial storage and computational resources to calculate diversity scores as $t$ continues to increase. To improve the long-term evolution efficiency, we propose a gradual manner where only the newly proposed instruction data $\mathbb{X}_{t+1}$ along with the data participated in last round of evolution $\mathbb{X}_t + \mathbb{B}_{t-1}^m$ are involved into the current round of evolution, and the evolution can be represented as $\mathbb{B}_{t+1}^m = \pi(\mathbb{X}_{t+1}, \mathbb{X}_t + \mathbb{B}_{t-1}^m)$.

In addition to the update of InsBank, we evaluate the diversity and quality of each sample $x_i$ and provide an overall individual score for data ranking. Users can quickly select a smaller subset according to the data ranking to suit their own training budget.

## 3.2 HISTORICAL INFORMATION FLOWING

Although we eliminate the vast amount of filtered-out data for long-term evolution efficiency during InsBank evolution, the distribution information of these data should not be neglected to ensure the strong global representativeness of InsBank. In this work, we address this issue by maintaining a history information matrix that preserves the distribution information of the filtered-out data. With the history information flowing across the InsBank evolution iterations, the filtered-out data can be engaged into the future exemplar election, thus prevent the representativeness of the evolved InsBank being globally suboptimal.

As mentioned in Section 2.3, we adopt the affinity propagation framework to characterize individual diversity (i.e., the representativeness of each sample) whose values illustrate the suitability of one sample to serve as the exemplar for other samples, which can be viewed as the representation voting result indicating the data distribution characteristics. Utilizing the similarity between data from last round and newly proposed candidate data, we can further estimate the suitability of the new data to serve as the exemplars for previous selected data, and the suitability of previous selected data to serve as the exemplars for the new data. Similarity, the availability values also can indicate the data distribution characteristics. In practice, we only estimate the responsibility matrix since the calculation of availability matrix is based on the responsibility matrix.

Formally, let $\mathbb{X}'_t = \mathbb{X}_t \cup \mathbb{B}_{\pi}^{t-1,m}$ denote the full candidate data set from the previous round of InsBank evolution, where $\mathbb{X}_t$ represents the newly proposed candidate data in the $t$-th round, and $\mathbb{B}_{\pi}^{t-1,m}$ is the selected data in the $t$-th round. Let $\mathbb{X}_{t+1}$ denote the current newly proposed candidate data, then the full candidate data set of the $(t+1)$-th evolution round can be denoted as $\mathbb{X}'_{t+1} = \mathbb{X}_{t+1} \cup \mathbb{B}_{\pi}^{t,m}$. The matrix $Sim_{t+1}$ of size $|\mathbb{X}'_t| \times |\mathbb{X}_{t+1}|$ represents the cosine similarity between $\mathbb{X}'_t$ and $\mathbb{X}_{t+1}$.

Given history information matrix $R_t$ of size $|\mathbb{X}'_t| \times |\mathbb{X}'_t|$, which is the stored responsibility matrix from the $t$-th round of InsBank evolution, we estimate the momentum responsibility matrix $H_t$ from $R_t$ and $Sim_{t+1}$ to engage the filtered-out data to elect their own exemplars in the future history-aware AP process.

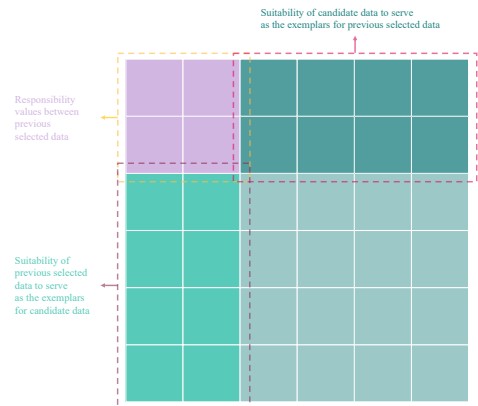

Figure 3: The structure of historical information.

Figure 3 depicts the structure of $H_t$. The top-left part of $H_t$ is filled by the responsibility values between data in $\mathbb{B}_{\pi}^{t,m}$ that are taken from $R_t$ directly, and the bottom right part of $H_t$ is filled by 0. The top right part of $H_t$ illustrates the suitability of newly proposed candidate data to serve as the exemplars for previous selected data, and the values of this part can be estimated by Eq. 4:

$$w_{jk} = Sim[j,k] * \frac{Sim[j,k]}{\sum_{l=1}^{|X'_t|} Sim[l,k]},$$

$$H_t[i,k] = \sum_{j=1}^{|X'_t|} w_{jk} * R_t[i,j]$$

(4)

Similarly, the bottom left part of $H_t$ illustrates the suitability of previous selected data to serve as the exemplars for newly proposed candidate data, and the values of this part can be estimated by Eq. 5:

$$H_t[i, k] = \sum_{j=1}^{|X'_t|} w_{ij} * R_t[j, k] \tag{5}$$

However, since part of values in $R_t$ can be negative and $\sum_{j=1}^{|X'_t|} w_{jk}$ is always less than 1, the estimated values in the top-right of $H_t$ can thus be greater than the values in the top-left. Additionally, by filling the bottom-right of $H_t$ with zeros, a significant preference for the new candidate data would be introduced to the momentum responsibility matrix. To tackle this issue, we add a correction value to the right half of $H_t$ which is the smaller of the median value of $R_t$ and 0.

We regard $H_t$ as a continuously decaying momentum term during the message passing iterations of the $(t + 1)$-th round of data selection. Specifically, we first calculate $R_{t+1}^i$ and $A_{t+1}^i$ according to Eq. 3 in the $i$-th message passing iteration. Then, we apply a weighted sum of $H_t$ and $R_{t+1}^i$ to mitigate the forgetting of historical information as shown in follows:

$$R_{t+1}^i = \alpha_i \cdot H_t + (1 - \alpha_i) \cdot (\beta \cdot R_{t+1}^i + (1 - \beta) \cdot R_{t+1}^{i-1}), \tag{6}$$

where $\alpha_i = \lambda \cdot \alpha_{i-1}$ is the momentum coefficient with a decay rate of $\lambda$, and $\beta$ is the damping rate to prevent numerical oscillations between iterations (Frey & Dueck, 2007). Both $\alpha$, $\lambda$ and $\beta$ are predefined hyperparameters. In this work, damping rate $\beta$ is set to 0.5 unless otherwise specified.

### 3.3 REPRESENTATION SCORING

The individual representation score encapsulates the exemplar election information, reflecting both the willingness of other samples to be represented by a specific sample and the unwillingness of the specific sample to be represented by samples. As mentioned above, the responsibility value $R[i, k]$ represents the suitability for $x_k$ to serve as the exemplar for $x_i$ and the availability value $A[i, k]$ represents the appropriateness for $x_i$ to select $x_k$ as its exemplar. By adding $A$ and $R$, $(A + R)[i, k]$ represents the combined evidence from $x_i$ to select $x_k$ to be its exemplar (Frey & Dueck, 2007). Thus, the sum of the $k$-th column of $(A + R)$ can be regarded as the fitness of $x_k$ to represent other samples, while the sum of the $k$-th row of $(A + R)$ represents the fitness of $x_k$ being represented by other samples. Subsequently, the representation score of $x_k$ can be obtained through Eq. 7:

$$s_{rep}^k = \sum_{i=1}^{|X'_{t+1}|} (A + R)[i, k] - \sum_{i=1}^{|X'_{t+1}|} (A + R)[k, i] + (A + R)[k, k] \tag{7}$$

### 3.4 INTEGRATION OF DIVERSITY AND QUALITY

Both quality and diversity of instruction data are of great importance for instruction tuning. However, existing data selection methods typically either only focus on one alone or handle them one after the other, without giving equal attention to both aspects simultaneously. Here, we combine the quality scores and diversity scores in two manners: addition and multiplication, both preceded by normalization. As illustrated in Eq. 8, we applied min-max normalization to the quality scores $s_q^k$ and diversity scores $s_{rep}^k$ to address the issue of their inconsistent scales.

$$s'_{rep}^k = \frac{s_{rep}^k - \min_{x_i \in X'_{t+1}} s_{rep}^i}{\max_{x_i \in X'_{t+1}} s_{rep}^i - \min_{x_i \in X'_{t+1}} s_{rep}^i}, \quad s'_q^k = \frac{s_q^k - \min_{x_i \in X'_{t+1}} s_q^i}{\max_{x_i \in X'_{t+1}} s_q^i - \min_{x_i \in X'_{t+1}} s_q^i} \tag{8}$$

In the addition manner, the normalized scores of quality and diversity are summed, providing a balanced approach that treats both aspects independently but equally. This method allows for flexibility when the contributions of quality and diversity are to be considered as separate yet additive factors.

$$s^k = s'_{rep}^k + \gamma \cdot s'_q^k. \tag{9}$$

In the multiplication manner, the normalized scores are multiplied, which emphasizes the interactions between quality and diversity. This approach is more sensitive to cases where either factor is low, ensuring that both quality and diversity must be sufficiently high to produce a high combined score[1].

$$s^k = (1 + s'^k_{rep}) * (1 + s'^k_q)^\gamma \tag{10}$$

Eq. 9 and Eq. 10 illustrate the calculation of the individual overall score using the additive and multiplicative approaches, respectively, where $\gamma$ is the weighting coefficient that controls the focus between diversity and quality.

After getting the overall scores, in addition to serving as the criterion for InsBank evolution, users can quickly select a smaller subset according to the data ranking to suit their own training budget.

# 4 EXPERIMENT

## 4.1 EXPERIMENTAL SETUP

Table 1: Statistics of instruction datasets.

| Dataset | Scale | Quality |
|---|---|---|
| Self-Instruct | 82k | 2.29 |
| Alpaca | 52k | 3.59 |
| Dolly | 15k | 2.76 |
| ShareGPT | 58k | 4.03 |
| WizardLM | 70k | 4.16 |

Table 2: The overlap rate between data selected from full-scale scenario and data selected in the progressive scenario.

| Method | Overlap Rate |
|---|---|
| kNN$_i$ | 131 |
| kCenter Greedy | 187 |
| PIBE w/o history | 681 |
| PIBE | 833 |

**Candidate Instruction Data** We aggregate Self-Instruct (Wang et al., 2023), Alpaca (GPT-4) (Peng et al., 2023), Dolly (Conover et al., 2023), ShareGPT[2] (Chiang et al., 2023) and WilzardLM (alpaca) (Xu et al., 2023) resulting in a mixed dataset of 278k samples. The statistics of each dataset is presented in Table 1.

**Training and Evaluation** In this work, we fine-tune the Llama3 8B model (AI@Meta, 2024) on the selected InsBank unless otherwise specified. Following DEITA (Liu et al., 2024), we adopt a data budget $m$ of 6K samples. The hyperparamters utilized during data selection and instruction fine-tuning can be found in Appendix A. For the evaluation, we employ AlpacaEval (Li et al., 2023b) and MT-Bench (Zheng et al., 2023) with GPT-4 turbo as the annotator.

**Baselines** We compare proposed PIBE with the following baselines:

- **kNN$_1$** Measure the diversity of one sample by its euclidean distance to the nearest neighbor (Eq. 2). The diversity score is first normalized and then combine with the normalized quality score by $s_i = (1 + kNN^i_1) * (1 + s'^i_q)^\gamma$ for data selection.
- **kCenter Greedy** (Sener & Savarese, 2018) The original kCenter Greedy algorithm is shown in Alg. 1. We take $\min_{x_j \in S_b} d(e(x_i), e(x_j))$ as the individual diversity score and combine it with quality score in the same manner of kNN$_1$.
- **DEITA** Traverse the instruction pool in descending order of quality scores and involve the current sample to the selected subset if the largest cosine similarity between the current sample and the samples in the selected subset is less than the threshold (i.e. 0.9 following the raw setting of DEITA (Liu et al., 2024)).

## 4.2 COMPARISON BETWEEN PROGRESSIVE EVOLVING AND FULL DATA SELECTION

In this section, we aim to compare the overlap rates between the subsets selected by different methods from the gradual manner and those from the full-scale selection manner [3]. We first randomly sampled

---

[1] In this work, we employ the multiplication manner to calculate overall score unless otherwise specified

[2] We filter out incomplete conversations.

[3] Aggregate all available candidates first and perform data selection on the full data directly.

Table 3: Comparison between different data selection approaches. For MT-Bench, we employ gpt-4o as the annotator. The bolded results indicate the best performance which significantly surpass other methods, while the underlined results represent the best performance, though without a significant advantage over other methods.

| Model | MT-Bench | MMLU | HellaSwag | ARC | TruthfulQA | Winogrande |
|---|---|---|---|---|---|---|
| $kNN_1$ | 5.93 | 0.64 | 0.82 | 0.62 | 0.55 | 0.75 |
| kCenter Greedy | 4.83 | 0.62 | 0.81 | 0.58 | 0.41 | 0.73 |
| DEITA | 5.96 | 0.64 | 0.82 | 0.60 | 0.54 | 0.74 |
| PIBE (ours) | **6.11** | 0.64 | 0.82 | 0.60 | 0.51 | 0.74 |

40k samples from the full candidate pool as a candidate set, and selected 1k samples from this set as the full-scale selection result. Next, we divided the data into four candidate subsets of 10k each to simulate the gradual manner. We compared PIBE with $kNN_1$ and k-Center Greedy, and performed an ablation analysis on the historical information used in PIBE. Here, we set $\gamma = 1$, $\alpha = 0.5$ and $\lambda = 0.5$. The size of InsBANK here is 1k. The results are reported in Table 2. It shows that the overlap rate of PIBE significantly exceeds that of the $kNN_1$ and kCenter Greedy, and that historical information also helps improve the overlap rate. This demonstrates that our proposed message-passing-based diversity representation method is highly robust for the gradual evolution of datasets and effectively leverages historical information.

Table 4: Results of progressive InsBank evolution. For each method, the $i$-th row from top to bottom shows the results of the model fine-tuned on InsBank obtained in the $i$-th evolution turn.

| Model | MT-Bench | AlpacaEval (GPT-3) | AlpacaEval (GPT-3.5) | AlpacaEval (GPT-4) |
|---|---|---|---|---|
| | 3.95 | 37.26 | 4.04 | 1.79 |
| | 6.01 | 86.16 | 33.21 | 5.25 |
| DEITA | 5.78 | 85.39 | 33.21 | 5.68 |
| | 6.71 | 90.43 | 41.33 | 8.31 |
| | 6.79 | 90.43 | 43.14 | 7.72 |
| | 3.85 | 37.97 | 5.09 | 1.54 |
| | 5.91 | 89.29 | 41.17 | 7.32 |
| PIBE (ours) | 5.70 | 87.94 | 41.59 | 6.85 |
| | 6.86 | 90.01 | **47.88** | 8.50 |
| | **6.93** | **91.21** | 47.58 | **10.70** |

## 4.3 Evaluating Progressive Instruction Bank Evolution

In this experiment, we investigate the performance of subsets selected by different data selection methods for model training. First, we applied $kNN_1$, k-Center Greedy, and DEITA for data selection on the full dataset. Next, following the temporal order of dataset appearance (i.e. Self-Instrucy $\rightarrow$ Alpaca $\rightarrow$ Dolly $\rightarrow$ ShareGPT $\rightarrow$ WizardLM (alpaca)), we performed progressive InsBank evolution using PIBE and take the final selected subset for model fine-tuning. The performance of the fine-tuned model across different benchmarks is shown in Table 3. The results indicate that our PIBE method outperforms all baselines, achieving a MT-Bench score of 6.93, a 91.21 win rate against GPT-3, a 47.58 win rate against GPT-3.5, and a 10.70 win rate against GPT-4 on AlpacaEval.

Further, we compare the performance of DEITA and PIBE under the progressive data selection scenario. In each round of subset evolution of PIBE, new instruction dataset is introduced, along with the previously selected 6k samples, to participate in the current data selection process. The results of model fine-tuned on each selected subsets are reported in Table 4. According to the results, both DEITA and PIBE succeed to evolve the selected subsets with the benchmark scores of the fine-tuned model improves in parallel of the development of instruction datasets. However, PIBE continues to outperform DEITA with higher benchmark scores in each evolution round, which demonstrates the superiority of PIBE.

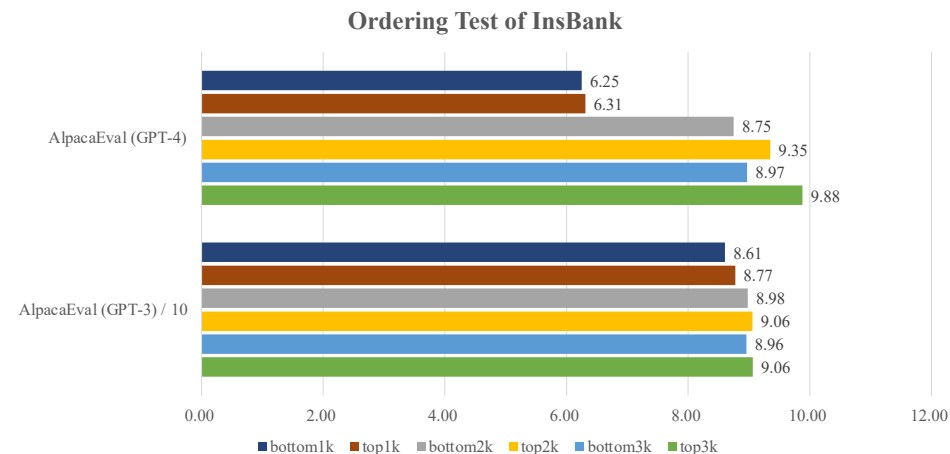

Figure 4: Results of the performance of models fine-tuned with the "top1k, bottom1k", "top2k, bottom2k", "top3k, bottom3k" samples in InsBank. In order to unify the scale of values, for AlpacaEval (GPT-3), we show the values after they are reduced by ten times

## 4.4 ORDERING OF INSBANK

Each sample in the InsBank selected by PIBE is provided with an overall individual score reflects both the diversity and quality which shows the priority of each sample to be used to fine-tune models. We sort the InsBank by the overall individual score, and compare the performance of models fine-tuned with the "top1k, bottom1k", "top2k, bottom2k", "top3k, bottom3k" samples in InsBank. Here, we use the instruction subset obtained from the final evolution round. We report the results on AlpacaEval against text-davinci-003 and gpt4-1106-preview which align with standard AlpacaEval1.0 and AlpacaEval2.0, where results are shown in Figure 4. In each comparison pair, the top samples always outperforms the bottom samples, demonstrating the effectiveness of our individual overall scores, and users can further select the top $b$ samples for instruction fine-tuning to suit their own training budget.

Table 5: Results of score combination analysis. The number in each method name refers to the $\gamma$ value it adopts.

| Method | AlpacaEval (GPT-3) | AlpacaEval (GPT-4) |
|---|---|---|
| Addition 1 | 91.18 | 9.21 |
| Addition 2 | 91.21 | 10.70 |
| Addition 3 | 89.18 | 10.43 |
| Multiplication 1 | 90.31 | 8.77 |
| Multiplication 2 | 89.93 | 9.67 |
| Multiplication 3 | 91.27 | 11.05 |

## 4.5 ANALYSIS OF SCORE COMBINATION

In this section, we experiment with the different combination methods for quality and diversity. We shows the results of the addition manner shown in Eq. 9 and the multiplication manner shown in Eq. 10 with different values of $\gamma$ to explore the contribution of quality and diversity in PIBE. The results are reported in Table 5. We find that the fine-tuned model performance rises first and then falls as the value of $\gamma$ increases for addition combination. This observation demonstrates that neither the diversity nor the quality dominates the training performance of InsBank. The multiplication manner exhibited a different trend, possibly because in this approach, larger coefficients reduce the likelihood of selecting low-quality data. As for exploring larger coefficients, we leave that for future work.

## 5 RELATED WORK

Instruction Fine-Tuning are widely adopted to stimulate the instruction following capability of pre-trained LLMs. Early approaches focused on fine-tuning LLMs with large amounts of instruction data (Wei et al., 2022; Wang et al., 2022) manually aggregated from large NLP task collections (Longpre et al., 2023). With the development of generative language models, Wang et al. (2023) made their attempt to expand instruction data through synthetic data generation, inspiring the following works to evolve instruction data in this automated manner (Taori et al., 2023; Ding et al., 2023; Xu et al., 2023). Zhou et al. (2023) proved that the quality and diversity of instruction data are significantly more critical than its sheer quantity, motivating recent efforts in instruction data selection to remove unnecessary training costs by eliminating low-quality and redundant data. Existing data selection methods can be systematically categorized into three types (Qin et al., 2024a):

**Quality-based Selection**   The quality of instruction data primarily refers to the accuracy and rationality which estimate the consistency and coherence of the instruction context, as well as whether the response accurately corresponds to the instructions. Humpback (Li et al., 2023a) selects high-quality samples through an iterative self-curation process where quality predictions are produced by the fine-tuned model of each turn. Recent works typically employ a GPT-model to annotate the data quality. For example, ALPAGASUS Chen et al. (2024) employs ChatGPT to score the accuracy of instruction data and select data according to a threshold.

**Diversity-based Selection**   The diversity-based selection aims to deduplicate the instruction data and maximize the coverage of selected data. Recent methods typically achieve this purpose by control the nearest neighbor distance (Liu et al., 2024) or maximize the average distance between the selected data through text embedding (Wu et al., 2023). INSTAG (Lu et al., 2024) identifies semantics and intentions of instructions by tags and it assumes that a dataset is considered more diverse if it covers more individual tags.

**Model-specific Importance-based Selection**   The importance refers to the necessity of adding one sample into the training set (Liu et al., 2024) whose indicator are typically model-specific (Xia et al., 2024; Li et al., 2024a). However, in this work we focus on the general data selection scenarios and emphasize the quality and diversity of selected data.

InfoGrowth (Qin et al., 2024b) also aims to address the continuous expansion of datasets, but it primarily deals with image data and focuses on relabeling noisy samples, making it less relevant to this paper. InfoGrowth and DEITA take both data quality and diversity into consideration. However, both of them handle the two aspects sequentially and fail to combine them together as an overall individual score. Besides, previous efforts primarily aggregate all candidate data first before performing data selection and are not experimented under the progressive instruction bank evolution task. While in this paper, we propose PIBE to efficiently obtain the optimal current instruction subset with comprehensive characterization and integration of diversity and quality scores.

## 6 CONCLUSION

In this paper, we introduced the concept of the Instruction Bank (**InsBank**) to address the ongoing challenge of evolving instruction datasets for large language models (LLMs). Our proposed Progressive Instruction Bank Evolution (**PIBE**) framework facilitates the effective and efficient evolution of InsBank by integrating new, high-quality instruction data while ensuring long-term scalability and efficiency. By leveraging a representation-based diversity score that retains historical information, PIBE provides a robust mechanism for comprehensive diversity evaluation, enabling the flexible combination of diversity and quality scores in the process of data selection and ranking. Extensive experimental results demonstrate that PIBE significantly outperforms baseline methods, providing more optimal and adaptable instruction subsets. Furthermore, the framework's flexibility allows users to extract smaller, tailored subsets based on their specific budget constraints, contributing not only to cost-effective training but also to the ongoing refinement and alignment of LLMs. This approach paves the way for more dynamic and adaptable instruction tuning strategies, enhancing both the efficiency and effectiveness of LLM development over time.

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

## A  DETAILS OF IMPLEMENTATION

**Fine-grained Quality and Complexity Scoring** We adopt the quality predictor [4] provided by Liu et al. (2024) to score the instructions.

**Representation-based Progressive Data Selection:** During the PIBE data selection process, we set the momentum coefficient $\alpha = 0.5$, the momentum decaying rate $\lambda = 0.99$, the damping rate $\beta = 0.5$ and the weighting coefficient $\gamma = 1$. Besides, we adopt instruction embedding (Li et al., 2024b) to encode the instructions. As for affinity propagation, we use negative euclidean distance to initialize the similarity matrix and fill the diagonal of similarity matrix with 0.

**Instruction Fine-Tuning:** We utilize 8 NVIDIA A100 SXM4 40GB GPUs to fine-tune Llama3 8B model. We employ LlamaFactory (Zheng et al., 2024), DeepSpeed Zero-Stage 3 (Ren et al., 2021) and fp16 precision to facilitate the training process. We adopt the Llama3-style template, and set the effective batch size to 128 (per device train batch size=1 and gradient accumulation steps=16), training epochs to 6, learning rate to 1e-5, warmup ratio to 0.1 and maximum input length to 2048.

**Evaluation:** For AlpacaEval inference, we set temperature=0.7, top_p=0.9, top_k=40, num beams=1 and max length=512. For MT-Bench inference, we follow the default setting of FastChat[5] except for that max length is set to 512. Llama3 template is applied to both AlpacaEval inference and MT-Bench inference. For Open LLM LeaderBoard evaluation, we adopt the code of LM Evaluation Harness(Gao et al., 2024)[6] and follow the setting from `https://huggingface.co/spaces/open-llm-leaderboard-old/open_llm_leaderboard`.

## B  K-CENTER GREEDY

---

**Algorithm 1** K-Center Greedy

---

**Require:** data $x_i \in S$ and a budget $m$
1: Initialize $S_m = x_0$
2: **repeat**
3:     $u = \arg\max_{x_i \in S \setminus S_m} \min_{x_j \in S_m} d(g(x_i), g(x_j))$
4:     $S_m = S_m \cup \{u\}$
5: **until** $|S_m| = m$
6: **return** $S_m$

---

## C  VISUALIZATION

---

[4] `https://huggingface.co/hkust-nlp/deita-quality-scorer`
[5] `https://github.com/lm-sys/FastChat/tree/main`
[6] `https://github.com/EleutherAI/lm-evaluation-harness/tree/main`

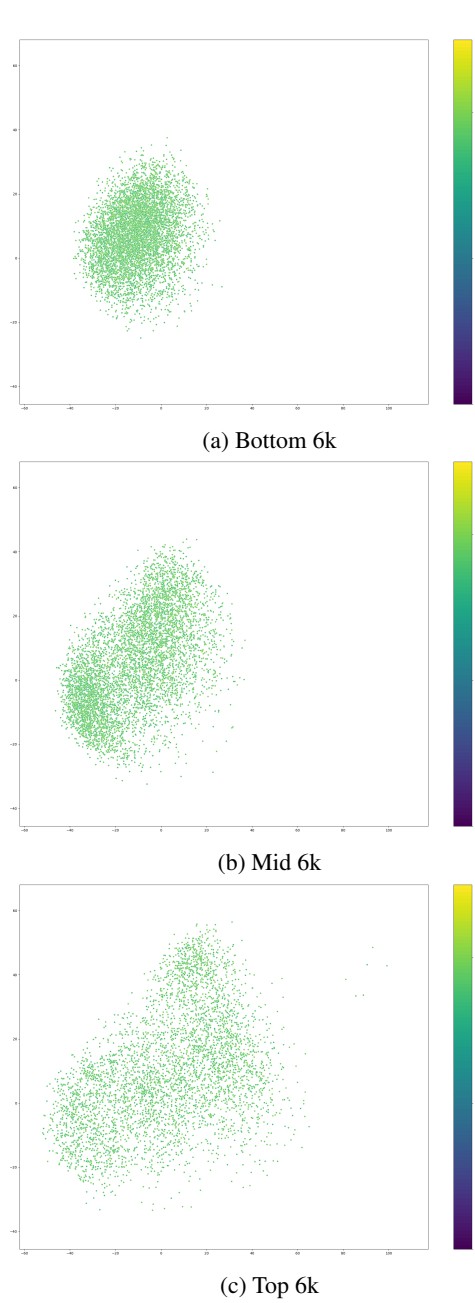

(a) Bottom 6k

(b) Mid 6k

(c) Top 6k

Figure 5: Visualization of subset selected according to representation score. Here, "top 6k" refers to the top 6k data points with the highest representation scores"bottom 6k" refers to the bottom 6k data points with the lowest representation scores, and "mid 6k" refers to the 6k data points around the middle range of the representation scores.

