# OpenReview forum: "InsBank: Evolving Instruction Subset for Ongoing Alignment"
_ICLR.cc/2025/Conference — ICLR 2025 Conference Withdrawn Submission_

### Official Review · Reviewer_QRuh · 2024-10-30

**Soundness:** 3
**Presentation:** 3
**Contribution:** 3
**Rating:** 6
**Confidence:** 3

**Summary:**

The manuscript introduces InsBank, a progressive instruction data repository, and PIBE, a framework for dynamically evolving instruction subsets. InsBank enables LLMs to continuously integrate new, high-quality, diverse instruction data for improved alignment and performance over time. Through abundant experiments, the authors proved that PIBE has significant advantages over the baseline method in the evolution of subsets of instructions.

**Strengths:**

1.The introduction of InsBank and the PIBE framework brings a novel solution to the ongoing alignment and evolution of instruction data for LLMs. I think it's a relatively comprehensive and novel framework
2.The adaptation of Affinity Propagation for diversity scoring is well-suited for this progressive approach, enhancing the robustness and representation quality of selected subsets.
3.The authors flexibly integrated quality and diversity scores, allowing PIBE to adapt to various budget constraints and maintain subset relevance over time.

**Weaknesses:**

1.The authors focus primarily on widely used datasets. I think it would be possible to evaluate the performance of PIBE on more domain-specific datasets or to evaluate its performance with multiple evaluation methods.
2.The ensemble weights for mass and diversity are not well analyzed, which can lead to issues with the sensitivity of PIBE performance to changes in these parameters.

**Questions:**

Please check the pros and cons of the paper which have been shown in the above list, I think this is a good paper.

---

### Official Review · Reviewer_dVeP · 2024-11-03

**Soundness:** 2
**Presentation:** 2
**Contribution:** 2
**Rating:** 3
**Confidence:** 3

**Summary:**

This paper addresses the challenge of selecting a diverse and high-quality instruction subset to enhance efficiency in instruction tuning. To achieve this, the authors score data points based on diversity and quality, using an affinity-propagation-based function for diversity scoring. In experiments, they evaluated their method, PIBE, against three baseline methods on two benchmark datasets.
The reviewer primarily has the following concerns regarding the significance of the problem, the problem formulation, the contribution, the presentation, and the experiments.

**Significance of the problem**

To the reviewer, the importance of selecting a subset of data for instruction tuning is not clear. From an efficiency perspective, considering the substantial data size involved in pre-training, the reviewer does not consider the instruction data size as a primary bottleneck hindering the development of foundation models. From a performance perspective, the authors did not provide sufficient evidence to demonstrate the benefits of data selection.

**Problem formulation**

As highlighted in numerous recent publications [1,2,3], instruction tuning is extensively used, in addition to alignment, to adapt LLMs for specific domains or tasks. For this reason, it would be critical to incorporate domain or task information into the data selection process, rather than using a task-agnostic approach as in the developed method.

**Contribution**

The contribution of this work is unclear. The challenges addressed don’t appear to be significant, as the main improvement seems to be an advanced clustering method over KNN for diversity measurement. This may be incremental and insufficient for a top-tier conference like ICLR.

**Presentation**

The presentation could be significantly improved. The motivations behind several key design choices are unclear. For instance, the advantages of affinity propagation over KNN for measuring data diversity are not clear. Additionally, the correlation between diversity measurement and model performance on downstream tasks is unclear. The rationale for calculating the representation score as in Eqn. 8 also needs clarification. Lastly, in Eqn. 4, please correct the font type for X and B on the right-hand side.

**Experiments**

In the experiments, the authors evaluate only three baseline methods and two benchmark datasets. Compared to similar studies in ICLR, the experimental setup lacks comprehensiveness. Additionally, it would strengthen the work if the authors reported the percentage of data selected and provided a comparison between using all data versus only the selected data, to better validate the effectiveness of the proposed method. It is also recommended that the authors demonstrate that their method enhances the diversity of the selected data and that the performance gains are primarily due to this diversity improvement.

[1] LlaSMol: Advancing Large Language Models for Chemistry with a Large-Scale, Comprehensive, High-Quality Instruction Tuning Dataset

[2] EcomGPT: Instruction-tuning Large Language Models with Chain-of-Task Tasks for E-commerce

[3] MAmmoTH: Building Math Generalist Models through Hybrid Instruction Tuning

**Strengths:**

* The developed method demonstrates superior performance over the considered baseline.

* The idea of using affinity propagation for diversity measuring is interesting

**Weaknesses:**

* This paper has weaknesses in problem formulation, contribution, presentation, and experimental design. Please see the summary for details.

**Questions:**

* What are the advantages of affinity propagation over KNN in diversity measuring?

* The authors are suggested to provide empirical or theoretical evidence on the improvement of diversity.

---

### Official Review · Reviewer_F7ry · 2024-11-04

**Soundness:** 3
**Presentation:** 3
**Contribution:** 3
**Rating:** 6
**Confidence:** 3

**Summary:**

To address the need for continuous alignment of LLMs with high-quality, diverse instruction data, this study introduces Instruction Bank (InsBank), a dynamic repository that continuously integrates valuable new instructional data. The authors propose Progressive Instruction Bank Evolution (PIBE), a framework designed to evolve InsBank efficiently by gradually selecting data based on a diversity score that considers both relationships among data points and historical diversity. This approach allows flexible combinations of diversity and quality scores for data selection, enabling customized, budget-conscious subset extraction. Experiments demonstrate that PIBE effectively enhances InsBank evolution, outperforming traditional methods.

**Strengths:**

1. The author consider an insteresting setting of contunually integrate instruction data selection for LLMs.

2. The prosposed method achieves a good performance on AlphacaEval and MT-Bench benchmarks.

**Weaknesses:**

1. The downstream evaluation benchmarks are limited. It would be better if the author conduct more downstream analysis on more benchmarks such as MMLU etc. to showcase the advantage of proposed method.

**Questions:**

Please refer to Weaknesses

---

### Official Review · Reviewer_JMsk · 2024-11-09

**Soundness:** 2
**Presentation:** 3
**Contribution:** 2
**Rating:** 5
**Confidence:** 4

**Summary:**

The paper introduces Instruction Bank (InsBank), a continuously updated repository that integrates the latest valuable instruction data to enhance the alignment of Large Language Models (LLMs) over time. Recognizing that the quality and diversity of instruction data are more critical than quantity, the authors address the challenge of evolving selected instruction subsets in tandem with new instruction data—a problem that has been insufficiently explored.

To tackle this, they propose the Progressive Instruction Bank Evolution (PIBE) framework. PIBE employs a gradual data selection strategy that maintains long-term efficiency by:

Utilizing a representation-based diversity score that captures relationships between data points.
Retaining historical information for comprehensive diversity evaluation.
Allowing flexible combination of diversity and quality scores during data selection and ranking.

**Strengths:**

Innovation in Data Management: The concept of InsBank and the PIBE framework addresses a critical need for efficient, ongoing alignment of LLMs with evolving instruction data.

Efficiency and Scalability: By retaining only necessary data and historical information, PIBE reduces computational and storage costs, making it suitable for large-scale applications.

Comprehensive Diversity Evaluation: The representation-based diversity score effectively captures relationships between data points, improving the quality of the selected subsets.

Flexibility: Users can adjust the balance between diversity and quality and select subsets that fit their specific training budgets.
Experimental Results: The framework's superiority over baseline methods on standard benchmarks.

**Weaknesses:**

Lack of novelty: While the paper presents the InsBank concept and the PIBE framework, the methods employed largely combine existing techniques without substantial innovation. The use of Affinity Propagation for diversity scoring and simple mathematical operations (addition and multiplication) to combine diversity and quality scores are straightforward applications of known methods.
Clarity in Methodology: need more detailed explanations of the experiments to enable result reproducibility.

Clarity in Methodology: need more detailed explanations of the experiments to enable result reproducibility.

Computational Complexity Analysis: A deeper analysis of the computational complexity of PIBE compared to other methods would strengthen the paper, especially regarding scalability to extremely large datasets.

**Questions:**

Parameter Sensitivity: How sensitive is PIBE's performance to the choice of hyperparameters like the momentum coefficient (α) and damping rate (β)? Is there guidance on how to select these parameters?

---

### Note · Authors · 2024-12-08

I have read and agree with the venue's withdrawal policy on behalf of myself and my co-authors.